# *miR-7* Controls the Dopaminergic/Oligodendroglial Fate through Wnt/β-catenin Signaling Regulation

**DOI:** 10.3390/cells9030711

**Published:** 2020-03-13

**Authors:** Lavanya Adusumilli, Nicola Facchinello, Cathleen Teh, Giorgia Busolin, Minh TN Le, Henry Yang, Giorgia Beffagna, Stefano Campanaro, Wai Leong Tam, Francesco Argenton, Bing Lim, Vladimir Korzh, Natascia Tiso

**Affiliations:** 1Genome Institute of Singapore, A-STAR, Singapore 138672, Singapore; lavanyaadusumilli21@gmail.com (L.A.); csiyangh@nus.edu.sg (H.Y.); tamwl@gis.a-star.edu.sg (W.L.T.); 2Department of Biology, University of Padova, 35131 Padova, Italy; nicola.facchinello@unipd.it (N.F.); giorgia.busolin@gmail.com (G.B.); giorgia.beffagna@unipd.it (G.B.); stefano.campanaro@unipd.it (S.C.); francesco.argenton@unipd.it (F.A.); 3Institute of Molecular and Cell Biology, A-STAR, Singapore 138632, Singapore; dbstc@nus.edu.sg; 4Department of Biological Sciences, National University of Singapore, Singapore 117558, Singapore; 5Department of Pharmacology, National University of Singapore, Singapore 117559, Singapore; phcltnm@nus.edu.sg; 6International Institute of Molecular and Cell Biology, 02-109 Warsaw, Poland

**Keywords:** *miR-7*, zebrafish, Wnt, Shh, Tcf, dopaminergic neurons, oligodendrocytes, glia

## Abstract

During the development of the central nervous system, the proliferation of neural progenitors and differentiation of neurons and glia are tightly regulated by different transcription factors and signaling cascades, such as the Wnt and Shh pathways. This process takes place in cooperation with several microRNAs, some of which evolutionarily conserved in vertebrates, from teleosts to mammals. We focused our attention on *miR-7*, as its role in the regulation of cell signaling during neural development is still unclear. Specifically, we used human stem cell cultures and whole zebrafish embryos to study, in vitro and in vivo, the role of *miR-7* in the development of dopaminergic (DA) neurons, a cell type primarily affected in Parkinson’s disease. We demonstrated that the zebrafish homologue of *miR-7* (*miR-7a*) is expressed in the forebrain during the development of DA neurons. Moreover, we identified 143 target genes downregulated by *miR-7*, including the neural fate markers TCF4 and TCF12, as well as the Wnt pathway effector TCF7L2. We then demonstrated that *miR-7* negatively regulates the proliferation of DA-progenitors by inhibiting Wnt/β-catenin signaling in zebrafish embryos. In parallel, *miR-7* positively regulates Shh signaling, thus controlling the balance between oligodendroglial and DA neuronal cell fates. In summary, this study identifies a new molecular cross-talk between Wnt and Shh signaling pathways during the development of DA-neurons. Being mediated by a microRNA, this mechanism represents a promising target in cell differentiation therapies for Parkinson’s disease.

## 1. Introduction

Parkinson’s disease (PD) is one of the most common conditions affecting the elderly population. It causes disabling motor abnormalities that manifest as tremor, slow movements, rigidity, and poor balance. These impairments stem from the progressive loss of dopaminergic (DA) neurons in the substantia nigra pars compacta. Later, many patients develop dementia and hallucinations owing to defects in other brain regions. 

Therefore, one of the main challenges in PD is the identification of the developmental mechanisms whose modulation can promote DA neuron formation, in view of cell differentiation therapies for PD.

It is well known that the induction and maintenance of the midbrain DA neurons (mDA) rely on several signaling molecules including Shh, FGF, TGFβ, and Wnt [1]. For instance, SHH, generated by the notochord and floor plate, and FGF8, by the isthmus, participate in the induction of mDA neurons; TGFβ signaling was also found to play an important role in this process [2]. Several transcription factors are involved in neural lineage specification and mDA neuron development [3,4]. Among these, LMX1A is an early inducer of DA neurons through MSX1, which in turn promotes NGN2 activation, eventually resulting in differentiation of tyrosine hydroxylase-positive (TH+) DA neurons. The DA progenitor profile in the ventral midbrain is defined by a combination of LMX1A/MSX1/NKX6.1 expression, while factors such as LMX1B, PITX3, NURR1, and EN1 are present in mature neurons. Moreover, PITX3 is specific for mature DA neurons expressing the TH marker in the substantia nigra pars compacta in mammals or the posterior tuberculum of the zebrafish brain [5]. Finally, the E proteins TCF4 and TCF12, belonging to class A of bHLH (basic helix–loop–helix) transcription factors, participate by interacting as heterodimers with other bHLH proteins, in this way forming combinations with unique specificity in neural lineage specification [6].

Wnts are secreted signaling proteins implicated in many conserved aspects of vertebrate neural development, from fish to humans [7,8]. Wnt pathway activation eventually results in the formation of the β-catenin-T-cell specific transcription factor (TCF) complex, which, binding the TCF consensus motif AGATCAAAGG, activates the transcription of Wnt target genes. Broadly speaking, Wnt signals have influential roles in controlling cell proliferation, cell fate determination, and terminal differentiation of post-mitotic cells [7]. During neural development, Wnt signaling has been implicated in the formation of mDA neurons and supporting glia cells, such as myelinating oligodendrocytes, regulated by the transcription factor Olig2 [9,10,11,12,13]. Interestingly, one of the TCF proteins (Tcf7l2) forms a complex with Olig2 during remyelination [14], and has been shown to be involved in forebrain neuronal specification [15].

Recent evidence has begun to implicate some miRNAs in neural development as well as in neurological disorders, including PD, where they act as key regulators of the above-mentioned signaling pathways and transcription factors [16]. MicroRNAs (also referred to as miRNAs or miRs) are small non-coding RNA molecules playing a key role in RNA silencing and post-transcriptional gene regulation. Many miRNAs are expressed in a tissue-specific manner and their functions can be attributed to the mRNAs that they regulate in each tissue. miRNAs may play important roles in the control of many different physiological and pathological functions, including stem cell self-renewal, cell differentiation, cancer, and neurodegeneration [17]. *miR-133b*, for example, is a midbrain-specific miRNA found to be deficient in PD midbrain tissue and is involved in the maturation of DA neurons through a negative regulatory feedback loop involving the transcription factor PITX3 [17].

More recently, another miRNA, *miR-7* (also known as *miR-7a*), has been shown to spatially control the rate of DA neuron production in the forebrain, by post-transcriptionally regulating the DA determinant Pax6, a known Wnt-target factor [18]. *miR-7* has particularly attracted our attention, as this is a highly-conserved miRNA, from annelids to humans [19], whose role in the normal development of DA neurons and other neural cells is still unclear. Further, how its activity relates to brain-activated signaling pathways is not yet an investigated aspect.

To gain further insight on *miR-7* neural activity, we applied an experimental approach based on the comparative analysis of human cell differentiation and zebrafish embryonic development upon *miR-7* perturbation. The zebrafish organism lacks a midbrain DA system; however, it possesses an ascending DA system in the ventral diencephalon and shares an evolutionary conserved set of DA markers [20]. We report here on the expressional and functional analysis of *miR-7*, which we found to be upregulated during human neural differentiation *in vitro*, and which is specifically expressed in the zebrafish ventral brain. 

Interestingly, among 143 identified targets for *miR-7*, we detected the genes encoding the bHLH neural fate markers *TCF12* and *TCF4,* as well as the TCF/LEF Wnt signaling-effector *TCF7L2*. 

Through over-expression and knock-down experiments in human cells and zebrafish carrying Wnt pathway reporters, we demonstrate here that *miR-7* negatively regulates the Wnt/β-catenin response, playing a key role in the balance between oligodendroglial and DA neuronal cell fates.

## 2. Materials and Methods

### 2.1. Cell Culture Conditions

H9 is a pluripotent human ESC line, representing an ideal system for differentiation studies. H9 cells (passages 25–35) were obtained from Dr. Lin Lin (Prof. Lawrence Stanton’s lab) and maintained on Matrigel coated plates in mTESR medium under feeder free conditions. HEK293T is a cell line derived from differentiating embryonic kidney, suitable for transfection and TOP/FOP flash assays (see later in this section). HEK293T cells were obtained from ATCC and maintained in DMEM medium supplemented with 10% fetal bovine serum, 1% L-glutamine, 1% sodium pyruvate, and 1% penstrep.

### 2.2. Neural Induction and Differentiation

H9 cells at about 20% confluency were treated with 4 μM CHIR99021 (GSK3 inhibitor, Cellagentech, San Diego, CA, USA), 3 μM SB431542 (TGFβ signaling inhibitor, Cellagentech, San Diego, CA, USA), and 0.1 μM compound E (γ-Secretase Inhibitor XXI, Millipore, Singapore) in neural induction medium containing advanced DMEMF12/Neurobasal medium (1:1) 1×N2, 1×B27, 1% glutamax, 5 μg/mL BSA, and 10 ng/mL hLIF (Lifetech, Shenzhen, China) for seven days. The culture was then split 1:3 for the next six passages using Accutase and cultured in neural induction media supplemented with 3 μM CHIR99021 and 2 μM SB431542 on Matrigel coated plates; in addition, bFGF (20 ng/mL) and EGF (20 ng/mL) were added to sustain the proliferation of cells.

Spontaneous differentiation from H9 ES derived NPC was performed in DMEM/F12/Neurobasal medium (1:1), 1×N2, 1×B27, 300 ng/mL cAMP (Sigma-Aldrich, Singapore), and 0.2 mM vitamin C (Sigma-Aldrich, Singapore) (referred to as differentiation media) on matrigel coated plates. For dopaminergic neuron differentiation, cells were first treated with 200 ng/mL SHH (C24II), 100 ng/mL FGF8b (both from PeproTech, London, UK), and 200 μM ascorbic acid in N2B27 differentiation media for seven days for initial patterning, and then with 20 ng/mL BDNF, 20 ng/mL GDNF, 1 ng/mL TGF-β3, and 0.2m M dibutyryl cyclic AMP (Sigma-Aldrich, Singapore) for another 14–21 days.

### 2.3. Transfection of microRNA Duplexes and Antisense Morpholino Oligomers

ReNVM cells (passage less than 20) and human NPCs (passage less than 10) were seeded at 100,000 cells/well on Matrigel coated plates. On the next day, using 4 μL of Lipofectamine RNAimax (Invitrogen, Singapore), according to the manufacturer’s instructions, the cells were transfected with one of the following RNA oligonucleotides at 50 nM or 80 nM final concentration: scrambled duplex (NCDP) (PremiR negative control #1, Ambion, Thermo Fisher Scientific, Singapore) and microRNA 7 (*miR-7*) duplex (7DP) (PremiR). miRNA antisense or control morpholino oligomers were transfected at 100 nM concentration. After 5 h, the transfection medium was replaced with fresh growth medium or neuronal differentiation medium or dopaminergic differentiation medium, depending on the experiment. Morpholino oligomer sequences, targeting immature or mature zebrafish *miR-7a* forms, were as follows:

Immature form *loop 7a* MO-1: TTGTTGTCAGAAAGCAGAAGAAACA

Immature form *loop 7a* MO-2: TGTTGTCAGTACTGATGACGTCACA

Immature form *loop 7a* MO-3: TTGTTGTTGGTTTTTGTTCATTTTC

Mature form *miR-7a* MO: ACAACAAAATCACTAGTCTTCCA

Control (mismatch) *miR-7a* MO: AgAACAtAATCAgTAGTgTTCgA (mismatched bases in lowercase).

### 2.4. Cripsr/Cas9-Mediated Gene Editing

To knock-out (KO) the zebrafish *miR-7a* locus, single guide RNA (sgRNA) target sequences were selected using two freely available CRISPR design prediction tools: the CHOPCHOP program (available at https://chopchop.rc.fas.harvard.edu), and the Breaking-Cas software (available at https://bioinfogp.cnb.csic.es/tools/breakingcas/). Three common top-scoring target sequences shared between these two programs were chosen as sgRNAs for the KO of miR-7a. The sgRNAs were synthesized by Synthego (CA, USA) and resuspended in TE buffer (final concentration: 100 μM).

sgRNA “guide Upstream” (gU): 5′-ACTAGTCTTCCACAGCGAATCGG-3′

sgRNA “guide Internal 1” (gI1): 5′-TCACAGTCTACCTCAGCGAGCGG-3′

sgRNA “guide Internal 2” (gI2): 5′-CACAGTCTACCTCAGCGAGCGGG-3′

Genomic DNA was extracted using a HotSHOT-based protocol from three dpf gene-edited larvae, to verify the presence of mutations and confirm the activity of the sgRNAs in the F0 generation. Specifically, genomic fragments at the target sites were amplified by PCR with 5x HOT FIREPol Blend Master Mix (Solis BioDyne, Tartu, Estonia) and locus-specific primers (F: 5′-TTTCTCCAGACACCAGCACT-3′; R: 5′- ATCACACACGACTCACCTGT-3′); ZFIN accession: ZFIN:ZDB-MIRNAG-071204-1; reference sequence: XR_001797616.2. PCR products (expected size in controls: 179 bp) were analyzed in ethidium bromide-stained 3% low EEO agarose gel (Fisher BioReagents, Milan, Italy).

### 2.5. RNA Extraction and qRT-PCR

RNA was extracted from cells or zebrafish embryos using Trizol reagent (Invitrogen, Singapore) and subsequently column-purified with RNeasy kits (Qiagen, Singapore); the miRNEASY kit was used to isolate the microRNA. For qRT-PCR of the microRNA, 100 ng of total RNA was reverse-transcribed using the TaqMan microRNA RT kit and subjected to TaqMan miRNA assay (Applied Biosystems, Singapore). For qRT-PCR of mRNAs, cDNA synthesis was performed with 1 μg of total RNA using the High Capacity cDNA RT kit (Applied Biosystems, Singapore). The expression of all other genes was analyzed by SYBR assay (Applied Biosystems, Singapore) and using ABI Prism 7900HT Sequence Detection System 2.2 (Applied Biosystems, Singapore)

### 2.6. Gene Expression Microarray and Data Analysis 

Total RNA was extracted as described above. A total of 750 μg of total RNA was reverse-transcribed, converted to cRNA, labeled, purified, and applied onto the human HT-12 v4 BeadChip kit according to the manufacturer’s instructions. First, the respective backgrounds were subtracted from all raw data using Bead Studio (Illumina, Singapore) and then normalized using the cross-correlation method. Subsequently, normalized data were processed for the identification of differentially expressed genes using log2 1.5-fold as the critical value for the mean of log2 n-fold changes in expression between 7DP samples and NCDP controls.

Genes that were differentially expressed two days after the transfection of 7DP were subjected to gene ontology (GO) analysis. The percentage of these genes classified into each GO process was compared with that of the whole genome. Statistically significant (*p* < 0.05) classes were selected. For the clustering of genes differentially expressed two days post transfection, normalized and log2-transformed data were subtracted from the mean values across all arrays. Hierarchical clustering was then performed for these processed data using average linkages.

### 2.7. Target Prediction 

The targets of *miR-7* were predicted by different methods: Targets Scan 6.2, mirBASE target, and microcosm (EBI, Singapore) using default parameters. The downregulated genes from microarray were compared with Target Scan 6.2 predictions and only the common genes with a Pct (probability of conserved targeting) value >0.1 were selected for further analysis.

### 2.8. Luciferase Reporter Assay 

miRNA response element (MRE) or the entire 3′UTR of mouse *Pax6,* human *TCF12*, and partial 3′UTR (containing the seed match) of human *TCF4* were cloned into psiCHECK2 vector (Promega, Singapore) between the *XhoI* and *NotI* sites immediately 3′ downstream from the *Renilla* luciferase gene. The top (sense) and bottom (antisense) strands of each MRE were designed to contain *XhoI* and *NotI* sites, respectively. After synthesis, these were annealed and ligated into the psiCHECK-2 vector. 

Ten nanograms of each psiCHECK2 final construct was co-transfected with 10 nM 7DP or NCDP into HEK-293T cells in a 96-well plate using Lipofectamine-2000 (Invitrogen, Singapore). After 48 h, the cell protein extract was obtained, and firefly luciferase and *Renilla* luciferase activities were measured with the dual-luciferase reporter system (Promega, Singapore), according to the manufacturer’s instructions. The *Renilla* luciferase reading was normalized to the firefly luciferase activity. All experiments were performed at least thrice with triplicates.

### 2.9. TOP Flash/FOP Flash Assay 

Both TOP flash and FOP flash plasmids were obtained from Addgene (Watertown, MA, USA) http://www.addgene.org/12456/ and http://www.addgene.org/12457/ links, respectively. The TOP flash or FOP flash plasmid at 100 ng concentration was co-transfected with 7DP (20 nM) or NCDP (20 nM) and pRL-TK (*Renilla* luciferase internal control) at 10 ng concentration. The cell extract was obtained at 48 h post transfection and the luciferase activity was measured. Luciferase activities were normalized to the pRL-TK plasmid. All experiments were performed at least thrice with triplicates.

### 2.10. Western Blot Assay

Cells were lysed in RIPA buffer (Pierce, Thermo Fisher Scientific, Singapore) with protease inhibitor. The lysates were centrifuged at 4 °C for twenty minutes. The supernatant containing the protein was transferred to another tube. The sample concentration was measured using the Bradford assay. The protein samples (40 μg) were separated using 4–8% gradient precast gel (Invitrogen, Singapore) and transferred to the methanol-activated PVDF membrane for two hours. The membrane was then blocked in TBST containing 7.5% milk and subsequently probed with anti-TCF7L2 antibody (1:1000, mouse monoclonal Ab, Santa Cruz, Dallas, TX, USA), and anti-β-actin (1:5000, Santa Cruz, Dallas, TX, USA). The primary antibodies were dissolved in PBST with 5% milk and incubated overnight at 4 °C on a shaker. The membrane was washed with TBST five times. The secondary HRP-conjugated antibody goat anti-mouse IgG-HRP (1:5000, Santa Cruz, Dallas, TX, USA) was used for one hour at room temperature. The membrane, after washes with TBST, was developed using ECL Prime Detection reagent (Amersham, Amersham, UK). The membrane was then imaged using the ChemiDoc system from Biorad (Singapore), or in the dark room using X-ray films.

### 2.11. Animal Care and Fish Lines 

Wild-type and transgenic zebrafish (*Danio rerio*) lines were maintained by standard protocols [21,22]. At the Zebrafish Centre of the University of Padova, zebrafish embryos and adults were raised, staged, and maintained under standard conditions [23,24]. Wild type lines used in this work included Tuebingen, Giotto, and Umbria strains [25]. The following zebrafish transgenic lines were used: the Wnt-responsive lines *Tg(7xTCF-Xla.Siam:GFP)^ia4^* and *Tg(7xTCF-Xla.Siam:mCherry)^ia5^*, here renamed *Wnt:GFP* and *Wnt:mCherry*, respectively [26,27]; the Shh-responsive lines *Tg(Gli-d:mCherry)* and *Tg(Gli-d:GFP)*, here renamed *Shh:mCherry* and *Shh:GFP*, respectively [28,29]; and the *olig2:GFP* and *olig2:DsRed* transgenic lines, kindly provided by Prof. Bruce Appel Lab (http://zfin.org/ZDB-LAB-020513-1).

### 2.12. Animal Welfare

All animal experiments in Padova were performed at the Zebrafish Facility of the University of Padova, Italy, in accordance with the European (EU 2010/63 directive) and Italian Legislations.

The experimental protocols were approved by the University of Padova Panel for Animal Welfare (OPBA), with the authorization number 407/2015-PR (to NT) from the Italian Ministry of Health. All animal experiments in Singapore were carried according to the regulations of Institutional Animal Care and Use Committee (Biological Resource Center of Biopolis, license no. 120787), which approved this study.

### 2.13. Microinjection in Zebrafish Embryos 

All injections were carried out at 1- to 2-cell stage with 5 nl of solution into each embryo. In the *miR-7a* knockdown experiments, 500 fmoles of *miR-7a* morpholino and 500 fmoles of mismatch morpholino were injected per embryo; co-injection of the oligomers targeting the immature (loop) forms (loop7a1/2/3/) was done at a concentration of 0.15 pmoles/embryo for each loop form.

MicroRNA 7 7DP and NCDP were injected at 1 pmole/μL (5 fmoles per embryo). All solutions were made in Danieau buffer (58 mM NaCl, 0.7 mM KCl, 0.4 mM MgSO4, 0.6 mM Ca(NO3)2, 5.0 mM HEPES pH 7.6) containing 1% Phenol Red. For rescue experiments, morpholino and microRNA duplex solutions were mixed together, maintaining the above-mentioned concentrations for each component. For Crispr/Cas9-mediated *miR-7a* gene editing, fertilized eggs were injected with 5 nl of an aqueous solution containing Danieau buffer and 1% Phenol Red (for controls), plus 2 μM Cas9 protein (New England Biolabs, Milan, Italy) and 30 μM of each gRNA (for gene-edited embryos). At least 50 eggs were injected for each experimental condition. Experiments were done in triplicate.

### 2.14. Chemical Treatments

For Wnt/β-catenin signaling modulation, zebrafish embryos were exposed to the Wnt agonist SB216763 (S3442, Sigma-Aldrich, Milan, Italy), at 100 μM final concentration, or to the Wnt inhibitor XAV939 (X3004, Sigma-Aldrich, Milan, Italy), at 5 μM final concentration. For Shh signaling inhibition, zebrafish embryos were exposed to cyclopamine (C4116, Sigma-Aldrich, Milan, Italy) at 100 μM final concentration. Controls were treated with carrier solution only. All treatments were performed from one to two days post-fertilization.

### 2.15. Whole Mount In Situ Hybridization (WISH)

Whole mount *in situ* hybridizations (WISH) with double-Dig-labeled *miR-7a* miRCURY LNA (locked nucleic acid) probe (Exiqon, Qiagen, Singapore)) on zebrafish embryos were performed essentially as described [30]. The hybridization mix was prepared by adding 20 pmol of *miR-7a* doubled-labeled LNA probe to every 1 mL of hybridization solution. The hybridization temperature used was 20 °C below the melting temperature of the *miR-7a* LNA probe. Optimal signal-to-noise ratio during color development was obtained by washing the embryos with 5× Tris-buffered saline containing 0.1% Tween 20 (TBST buffer) between color reactions.

For all other probes, WISH was performed according to [31]. Double staining was carried out according to [30], using Fast blue and Fast Red (Sigma-Aldrich, Milan, Italy). Both blue and red precipitates could be fluorescently visualized, being far-red and red emitting, respectively. For homogeneous comparison of staining levels, control embryos, recognized by tail tip amputation (performed post mortem), were co-stained with treated sibs in the same tube. At least 20 embryos per experimental condition were processed in each experiment. The following probes were used: reporter *mCherry* and *egfp* [26], *id3* [32], *shha* [33], *th* [34], *lmx1bb* and *nr4a2a* [20], and *olig2* and *sox10* [12].

### 2.16. Whole Mount Immune-Detection of Dopaminergic Neurons

Zebrafish larvae (3 dpf) were fixed in 4% PFA overnight. The fixative was then removed and replaced with 100% methanol. A series of 5 min rehydration steps followed, with a graded decrease of methanol in PBS, pH 7.4 buffer (75% methanol, 50% methanol, and 25% methanol series). This was followed by 2 × 5 min PBST washes (containing 0.1% Tween 20). Then, 1 μL of molecular grade proteinase K was added to every 1 mL of PBST (1 min for “deformed” larvae and 5 min for “normal” larvae) to prime the larvae for antibody entry to the brain. 

Post-fixation with 4% PFA for at least 20 min was performed to quench residual proteinase K activity. This was followed by 4 × 15 min washes with PBST; no pre-incubation of larvae in blocking solution was carried out (it interferes with the signal). All treated larvae were incubated in 1:500 dilution of rabbit polyclonal anti-TH antibody (primary antibody) in PBST solution, on a mixer, at 4 °C, overnight. This was followed by 2 × 20 min washes of PBST followed by an overnight incubation of anti-rabbit AlexaFluor488 (1:1000) antibodies in PBST, at 4 °C. After removing the secondary anti-rabbit AlexaFluor488 antibody, all larvae were washed once in PBST (20 min). Adjustment of any subsequent washes depended on the fluorescent signal versus noise level observed under a fluorescent stereomicroscope. Larval eyes were manually removed to reveal the pretectum behind the eyes (under a stereomicroscope) before mounting in 1% agarose for confocal imaging.

### 2.17. Image Acquisition and Microscope Settings 

Fluorescent and bright-field images of the transgenic embryos were acquired using the LSM510 confocal laser-scanning microscope (Carl Zeiss Vision, Singapore). Projection of image stacks was made by the Zeiss image browser. Image analysis and signal quantification were performed using the Volocity 6.0 software (PerkinElmer, Singapore). Images were then imported into Adobe Photoshop for cropping, resizing, and orientation. Contrast and brightness were adjusted equally for all images of the same figure. For imaging the immune-stained cell culture plates, the Zeiss fluorescent microscope was used. For live imaging of transgenic lines, embryos were tricaine-anesthetized and embedded in low melting 1% agarose (A9414, Sigma-Aldrich, Milan, Italy). For post mortem imaging of stained embryos, samples were flat-mounted in 87% glycerol/PBS. Standard imaging was performed with a Leica M165 stereomicroscope, equipped with a Leica DFC480 digital camera. Confocal imaging was performed with a Leica SP5 spectral system. Image analysis and signal quantification were performed using the “Measurements” option of the Volocity 6.0 software (Perkin Elmer, Milan, Italy). Virtual qualitative localization of brain markers was performed using gene expression data available in the ViBE-Z (Virtual Brain Explorer for Zebrafish) database [35]. Final figures were assembled using Adobe Photoshop CS2.

### 2.18. Statistical Analysis 

All experiments were performed at least thrice (in some cases, four times) with two biological replicates each time. Statistical analysis was performed to determine the significance of differences between the treated samples and the controls, where values resulted from luciferase reporter assay, qRT-PCR, Western blots, high content screening, cell counting, and fluorescence intensity. Pairwise comparisons were made by unpaired t-test, while multiple comparisons were analyzed by one-way analysis of variance (ANOVA) followed by Tukey’s test (GraphPad Prism V6.0 software, GraphPad Software, San Diego, CA, USA). In the charts, error bars display standard errors of the mean. Significant differences from controls are indicated by asterisks. Two-tailed *p*-values and correspondence between asterisks and significance levels are indicated in the figure legends.

### 2.19. Data Availability Statement

Raw data and materials are available on request. Gene expression data have been submitted to the Gene Expression Omnibus (GEO) database at the NCBI/NIH (National Center for Biotechnology Information/National Institutes of Health), with the GEO accession number GSE128011.

## 3. Results

### 3.1. Identification and Validation of miR-7a Target Genes Including TCF4, TCF12, and TCF7L2

In a microarray screen for miRNAs differentially expressed upon differentiation of a human neuroblastoma cell line (SH-SY5Y), we identified *miR-7* as one of the most upregulated miRNAs during the process of neuronal differentiation; this upregulation was validated using Northern blot analysis [36].

To identify the targets of *miR-7* in neuronal progenitor cells, we performed a microarray analysis of H9 ES-derived human neural progenitor cells transfected with *miR-7* duplex (7DP), compared with a control duplex (NCDP). On the basis of the microarray data, we identified a set of *miR-7* downstream effectors that are downregulated by *miR-7* overexpression and contain putative binding sites for *miR-7* predicted by TargetScan and MicroCosm (Appendix A, GEO entry GSE128011).

Specifically, we found that *miR-7* over-expression in human neural progenitor cells down-regulated bHLH genes *TCF4* and *TCF12*, both involved in neural development, as well as *TCF7L2,* encoding a Wnt/β-catenin pathway TCF/LEF effector [37,38]. The top hits TCF4 and TCF12 belong to the bHLH group of transcription factors, with TCF4 playing a role in the pontine nucleus development [39], and TCF12 involved in mDA specification [4] and neural stem cell proliferation [40]. The total context scores for *TCF4* and *TCF12,* predicted by TargetScan, were 0.59 and 0.69, respectively. Specifically, *miR-7* has a 7mer-m8 seed match for *TCF4* and an 8mer seed match for *TCF12,* conserved from zebrafish to humans, in the 3′ UTRs of their mRNAs (Figure 1A).

The Wnt effector TCF7L2 had a TargetScan score below 0.1; however, manual target scanning on updated sequence releases could identify canonical *miR-7* binding sites (7_A1) in the 5′ UTR of human and mouse mRNAs, as well as canonical *miR-7* binding sites (7_m8 and 7_A1) in the 5′ and 3′ UTR of zebrafish *tcf7l2* mRNA (Figure 1A). 

To validate gene expression changes upon *miR-7* over-expression, we focused our attention on the two top hits *TCF4* and *TCF12*, performing real-time PCR and detecting, on average, a 40% decrease of both mRNAs levels; the known *miR-7* target *PAX6* was used as positive control [19] (Figure 1B). To verify that *TCF4* and *TCF12* were direct targets of *miR-7*, we cloned the entire 3′UTR region of human *TCF12* and the partial 3′UTR region of *TCF4* downstream of the *Renilla* luciferase gene in the psiCHECK2 vector. A 70% repression of the luciferase activity was observed in HEK-293T cells using a *miR-7*-responsive construct (7-MRE, positive control), and a 40% reduction for both *TCF4* and *TCF12* (Figure 1C). This repression was abolished by introducing a 3-base mutation in the *miR-7* seed match region of *TCF4* (negative control).

After discovering canonical *miR-7* sites in the 5′UTR of human, mouse, and zebrafish *TCF7L2* mRNAs, we confirmed that *miR-7* could decrease the protein levels of this Wnt transducer. According to our Western blot analysis, the TCF7L2 protein levels were significantly decreased upon *miR-7* over-expression in a dose-dependent manner (Figure 1D,E). Conversely, TCF7L2 protein levels increased upon knockdown of *miR-7* using antisense oligomers.

Overall, these data validated our analysis for potential *miR-7* target genes, confirming that, in human neural progenitor cells, two key bHLH neural fate regulators (*TCF4* and *TCF12*) are direct targets of *miR-7*, and that the protein levels of the Wnt signaling transducer TCF7L2 are negatively regulated by *miR-7*.

### 3.2. Zebrafish miR-7a Shows Tissue-Specific Expression during Development and Adulthood

To functionally characterize *miR-7 in vivo*, we selected the zebrafish owing to the ease of growth and analysis. The sequence of *miR-7* is conserved between human, mouse (*miR-7a* form), and zebrafish (*miR-7a* form) (http://www.mirbase.org/). To ascertain the spatiotemporal expression of zebrafish *miR-7a* during development, we performed *in situ* hybridization using DIG-labelled LNA probes specific to the *miR-7a* sequence. The staining was performed at various stages from 24 h post fertilization (hpf) to 96 hpf at 12 h intervals for each time point. *miR-7a* was first weakly expressed in the brain at around 36 hpf (data not shown), with high intensity at around 48 hpf (Appendix A). Within the brain, *miR-7a* is highly restricted to specific regions of the forebrain, namely in the telencephalon, hypothalamus, and adenohypophysis. Partial overlap of *miR-7a* was observed with Wnt-responsive and *th*-positive cells located in the diencephalic area of the zebrafish brain (not shown). Additionally, we observed *miR-7a* expression in the pancreas, at 96 hpf (Appendix A). The highly-restricted expression of *miR-7a* was retained in the adult (two years of age) in the same regions (Appendix A). By quantifying *miR-7a* expression using a more sensitive technique (qRT-PCR), we showed that the expression initiates at 48 hpf and continues to increase until 7 dpf. 

In the adult, the expression was very high (around 800-fold) in the forebrain region, while the hindbrain showed little or no expression (Appendix A).

### 3.3. miR-7 is a Negative Regulator of the Wnt/β-catenin Pathway in Both Human Cells and Zebrafish Embryos

As we found a gene encoding a key Wnt/β-catenin signaling effector (*TCF7L2*) among the *miR-7* down-regulated genes, we checked if zebrafish *miR-7a* could play a role in Wnt signaling regulation *in vivo.* For this purpose, we exploited the zebrafish Wnt reporter transgenic line Tg*(7xTCF-Xla.Siam:GFP)^ia4^* [26] to detect Wnt/β-catenin signal responsivity upon *miR-7* over-expression and knockdown. Tg*(7xTCF-Xla.Siam:GFP)^ia4^* embryos were injected at the 1-cell stage with anti-*miR-7* morpholino (MO) and a mismatch MO, and the phenotype was analyzed at 42 hpf. 

*miR-7* knockdown resulted in an increase in GFP expression in the ventral part of the brain, indicating increased Wnt activity (Figure 2A,B). To further validate the role of *miR-7* in the regulation of Wnt signaling, we used the TOP flash/FOP flash system, a Wnt-reporter assay. HEK293T cells were transfected with TOP flash or FOP flash, pRL-TK (control *Renilla* luciferase) vectors, and NCDP or 7DP. After 7DP transfection, the luciferase activity from the TOP flash vector, containing seven intact TCF-binding sites, was repressed by almost 60%. On the contrary, there was no significant change in the activity of the FOP flash vector, which contains mutated TCF binding sites (Figure 2C). 

On the other hand, *miR-7* knockdown, by either injection of anti-mature *miR-7a* MO or co-injection of anti-pre-*miR-7* loop MOs, revealed an increase of Wnt activity in the diencephalon of the zebrafish embryo (Figure 3A–D,G; MO validation shown in Appendix A).

Importantly, we found that the increased Wnt signaling observed after *miR-7a* knock-down was pheno-copied by *miR-7a* gene knock-out, obtained by Crispr/Cas9 genome editing (Figure 4). 

Specifically, we used a combination of multiple gRNAs targeting the *miR-7a* gene (Figure 4A) to induce a locus-specific indel modification (Figure 4B), resulting in a decrease of endogenous *miR-7* expression (Figure 4C–E). Gene-edited embryos (“crispants”) in Wnt-reporter background displayed a raise of Wnt signaling response in the ventral brain, ranging from a 20% up to an 80% increase, compared with control levels, depending on the gRNA combination (Figure 4 F–H). 

Taken together, zebrafish *miR-7* knockdown and knock-out both lead to an increase in Wnt signaling levels.

In addition, we assessed the expression of *id3*, a marker known to be induced by Wnt/β-catenin signaling, playing a role in cell-cycle progression in neural progenitor cells [40,41]. Upon *miR-7* knockdown, we observed an increase in *id3* expression levels along with an increased number of Wnt-responsive cells in the posterior tuberculum and hypothalamus (Figure 3E,F,H,I). 

Taken together, these experiments show both *in vitro* and *in vivo*, in human- and zebrafish-based setups, that *miR-7* exerts a negative control on Wnt/β-catenin signaling activation.

As Wnt/β-catenin signaling plays a pivotal role in the proliferation of progenitor cells, we checked in the same Wnt reporter line whether *miR-7* over-expression and knockdown influenced cell proliferation in zebrafish. 

For this purpose, we used the anti-phosphohistone-H3 (PHH3) antibody, a known marker for mitotic cells [42], to evaluate changes in cell proliferation upon *miR-7* manipulation. Over-expression of *miR-7* resulted in a reduction of proliferating cells in 2 dpf embryos with less or no PHH3 staining, particularly in the diencephalon (Appendix A). 

On the contrary, knockdown of *miR-7a* resulted in an increased number of PHH3+ cells in the brain, associated with Wnt+ cells (Appendix A). Overall, these data suggest that *miR-7a*, by modulating the Wnt/β-catenin pathway, regulates cell proliferation in the Wnt-responsive regions of the zebrafish brain.

### 3.4. miR-7a Regulates the Development of Dopaminergic Neurons 

Our data showed that the knockdown of *miR-7a* results in the activation of Wnt-responsive cells in the diencephalic region, especially in the ventral posterior tuberculum and hypothalamus of zebrafish embryos. Notably, these regions contain DA neurons in zebrafish [5,43,44].

Previous reports indicated that *miR-7a* had a role in mature DA neurons, as it was found to regulate translation of α-synuclein [45], whose aggregation is a major cause of PD. In parallel, several reports have previously highlighted the role of Wnt/β-catenin signaling in DA neurons’ development [46,47]. As we found that *miR-7a* negatively regulates the Wnt signaling pathway, we wanted to check whether *miR-7a* could affect DA neurogenesis. As a preliminary approach, we used the Virtual brain explorer (Vibe-Z)-based software [35] to check the possible co-localization of Wnt-responsive cells with regions positive for tyrosine hydroxylase (TH), a known marker for mature DA neurons [43]. Vibe-Z automatically maps cell signaling and gene expression data to 3D reconstructions of the zebrafish brain, with single-cell resolution. Using this software, we observed that a subset of Wnt-responsive cells in the zebrafish brain appeared to co-localize with TH+ cells at 72 hpf (Figure 5A,B). By whole-mount *in situ* hybridization, we detected a partial overlap between TH expression and Wnt signaling activation (Figure 5C,D), which corroborated the *in silico* prediction. These results prompted us to verify the effects of *miR-7a* perturbation on the DA neuronal population. Anti-*miR-7a* MOs were injected into zebrafish embryos at 1-cell stage and the TH differentiation marker was analyzed by *in situ* hybridization at 3 dpf. The knock down of *miR-7a* caused a significant increase in the number of TH+ cells in the diencephalic region of the brain (Figure 6A–F’’). On the contrary, over-expression of *miR-7a* resulted in a decreased number of TH+ cells (Figure 6G–G’’; quantifications in H,I). These variations appeared to specifically affect *th*-positive neuronal populations located in the ventral diencephalon (where most of zebrafish DA neurons are located) and, to a lesser extent, in the dorsal diencephalon, as well as in the hindbrain region, where *th*-positive noradrenergic populations are located. 

In addition, to assess the role of *miR-7a* at earlier stages of DA neuron specification, we used *nr4a2a* and *lmx1bb* as neural markers previously associated to DA precursor areas. Specifically, *nr4a2a* is required for the development of *th*-positive DA neurons in the pretectum, preoptic area, and retina. Conversely, *lmx1bb* is expressed in a diencephalic territory that might contain ventral diencephalon DA precursors, as its knock-down affects diencephalic DA neuron formation (as well as hindbrain noradrenergic populations) [20]. Knockdown of *miR-7a* resulted in an increased expression of both *nr4a2a*- and *lmx1bb*-positive cells, while over-expression of the *miR-7* resulted in a decrease of cells positive for both markers (Appendix A). From these zebrafish experiments, we deduced that *miR-7a* can negatively control the production of several immature and mature diencephalic DA populations *in vivo.* To find out whether these observations, made in the zebrafish model, could have implications for human biology, we further studied the role of *miR-7* in the neurogenesis of DA cells in NPCs, derived from human ES cells.

### 3.5. Human miR-7 Regulates the Development of Human DA Neurons In Vitro 

To evaluate the role of *miR-7* in DA neural development *in vitro*, we used the H9 NPC differentiation system to obtain a population containing DA neurons [48]. For the first 10 days, H9 NPCs were treated with SHH and bFGF for the patterning of neural progenitors, and for the next 14 days, these were treated with BDNF, GDNF, ascorbic acid, dibutyryl cAMP, and TGFβ to trigger differentiation of progenitors into DA neurons. After 21 days, neurons expressed DA-specific markers PITX3, LMX1A, TH, and NURR1, as assessed by immunostaining and qPCR (Figure 7).

To alter *miR-7* expression during differentiation, 7DP, NCDP, *miR-7* antisense, and control antisense were transfected twice at 72 h intervals. At 10 days post-transfection, TH immunostaining along with TUJ1 (mature neuronal marker) staining was carried out. A decrease in the number of TH+ neurons (~15%) was detected upon *miR-7* over-expression, when compared with NCDP (Figure 7A–C), while the knockdown of *miR-7* resulted in an increased number of TH+ neurons (~20%) (Figure 7D,E; quantifications in F). Overall, our data from both zebrafish and human-based neural differentiation assays point to a suppressive role for *miR-7* in DA neurogenesis.

### 3.6. Over-Expression of miR-7 Results in an Increase in Shh-Responsive and olig2-Expressing Cells in Zebrafish Embryos

Taken together, our results suggest that an excess of *miR-7* reduces the number of DA neurons, and that *miR-7* deficiency increases the number of DA neurons. 

Interestingly, our preliminary data from ReNVM and H9 NPC differentiation show also an increased expression of glial markers (PDGFRα and Sox10) upon *miR-7* overexpression (L. Adusumilli, preliminary observations). It is known that Wnt signaling plays a key role in oligodendrocyte development [11]. On the other hand, Shh has been shown to directly induce expression of Olig2, and a Wnt-Shh antagonism is known to underlie the regulation of oligodendrocytes development [49]. Thus, we sought to investigate *miR-7* levels in Shh and Olig2 reporter backgrounds, taking advantage of available *Shh:GFP* and *olig2:EGFP* zebrafish transgenic lines (see Mat&Met section). The injection of 7DP in *Shh:GFP* and *olig2:EGFP* embryos resulted in an increase of Shh-responsive and *olig2*-expressing cells, when compared with the control; this took place in parallel with a decrease of both signals when *miR-7* was knocked-down (Figure 8A–J). 

The positive influence of *miR-7* on Shh responsivity and *olig2* expression was confirmed independently from the green or red version of their reporters (Appendix A). Interestingly, for the *olig2* marker, increase upon *miR-7* over-expression was also confirmed in the spinal cord, using both transgene and antisense riboprobe analysis (Appendix A; quantifications in K). 

Finally, similar increased or decreased expression was observed for another oligodendrocyte marker, *sox10*, upon *miR-7* over-expression or knock-down, respectively (time-window: 48–60 hpf; not shown). Overall, these observations point to a role for *miR-7* in the positive regulation of oligodendrocyte progenitors. The role of *miR-7* on neuronal/glial markers was also verified in *miR-7* crispants, confirming an increase in the number of *th*-positive cells (Appendix A) and a decrease of *olig2* expression (Appendix A). Interestingly, the physiological expression of Shha was not significantly decreased (Appendix A), suggesting that *miR-7* may affect the number of Shh-responsive cells downstream of a substantially intact Shh signal.

### 3.7. miR-7 Manipulation Affects the Balance of Wnt/Shh Signaling in Neural Populations

To demonstrate the specificity of the MO-mediated *miR-7* deficiency, we attempted to rescue the phenotype by co-injecting *miR-7a* MO along with 7DP in double transgenic lines simultaneously expressing *Wnt:mCherry* and *Shh:GFP* reporters. We observed that the *miR-7a* morphant phenotype was partially rescued by over-expression of 7DP (Appendix A). In parallel, the over-expression or knockdown alone elicited opposite effects in the reporter lines (Appendix A; quantifications in E,F). Whereas *miR-7a* morphants and over-expressed embryos developed a characteristic phenotype and less than 10% survived with normal morphology by 2 dpf, the co-injection of 7DP and anti-*miR-7a* MO induced an 80% rescue of the normal balance of expression of *Wnt:mCherry* and *Shh:GFP* reporters (Appendix A), supporting the specificity of 7DP and anti-*miR-7a* MO activities in Wnt and Shh signaling regulation.

As a final step, we pharmacologically modulated Wnt and Shh signaling in parallel with *miR-7* downregulation and overexpression. This set of experiments, summarized in Appendix A, further support the role of *miR-7* on *th*-positive cells through Wnt and Shh signal balancing. Indeed, the downregulation of *miR-7* under Wnt signaling inhibition, as well as the upregulation of *miR-7* under Wnt over-activation, can rescue the physiological number of *th*-positive cells (Appendix A). The same result is obtained by upregulating *miR-7* under Shh signaling inhibition (Appendix A).

In conclusion, our results obtained in human cells and in developing zebrafish demonstrated that several genes encoding transcription factors involved in neurogenesis could be regulated by *miR-7* during development of DA neurons. Specifically, we show that TCF4 and TCF12, two bHLH factors known to be involved in the initiation of neuronal differentiation, including DA neuron formation, are directly regulated by this miRNA. Moreover, protein levels of the Wnt signaling effector TCF7l2, known for its interaction with Olig2 during development of oligodendrocytes, and for the regulation of thalamic and habenular neuronal identity [15], inversely correlate with *miR-7* levels. 

Notably, all three genes, *TCF4*, *TCF12*, and *TCF7L2*, here identified as *miR-7* targets, are expressed in the zebrafish diencephalon (ZFIN database), the area where we observed the strongest effects of *miR-7* modulation on DA neuronal populations.

Although a role for selected miRNAs as well as for Wnt and Shh signaling has been previously recognized in neural fate decision and DA development [50,51], no study at the whole-animal level has so far demonstrated how the balance between these signals, and the relative amount of DA neurons, could depend on the levels of a single miRNA.

Our results show the following: (i) *miR-7* is expressed in the embryonic brain region, where forebrain DA neurons are formed; (ii) *miR-7* levels inversely correlate with Wnt-signaling responsiveness and cell proliferation within that region; (iii) *miR-7* levels directly correlate with Shh-signaling responsiveness and the number of *olig2*-positive cells; (iv) Wnt- and Shh-responsiveness is simultaneously modulated when *miR-7* levels are manipulated *in vivo*; and (v) markers for immature and mature DA neurons inversely correlate with *miR-7* levels.

As summarized in Figure 9, *miR-7* activity on Wnt signaling suppression and DA neuron regulation is strictly interlaced with Shh responsiveness and oligodendrocyte marker upregulation, determining an opposite and balancing effect on these pathways *in vivo*. This could be at least in part an indirect outcome, owing to the known antagonistic role of the pathway transducers β-catenin (Wnt) and Gli1 (Shh) in regulating TCFs and their downstream target genes [52].

On the other hand, a direct role of *miR-7* on oligodendrocyte specification and maintenance has been previously proposed, based, however, on *ex vivo* and *in vitro* studies [53].

The expression pattern of zebrafish *miR-7a* in the forebrain is consistent with its role in the development of DA cells located in this region; specifically, the knockdown of *miR-7a* resulted in an *in vivo* increase of the Wnt-reporter GFP expression in the diencephalon, suggestive that *miR-7* negatively regulates diencephalic Wnt signaling and that this effect might be mediated by the Wnt transducer Tcf7l2, a TCF factor that we found to be significantly down-regulated upon *miR-7* over-expression. 

Our analysis of *miR-7* activity adds important missing elements in the molecular mechanisms of Wnt signaling regulation *in vivo*. In fact, the *miR-7*-dependent negative regulation of diencephalic DA cells appears to occur in the proliferative areas of the Wnt-responsive regions of the zebrafish brain, where DA precursors form and differentiate. This raises interesting implications in the analysis of molecular mechanisms underlying PD development, offering new miRNAs/signaling pathways cross-talks specifically targetable for its treatment, in the framework of existing research strategies [54,55,56].

In summary, based on an outcome of gain-of-function and loss-of-function experiments, we conclude that *miR-7* acts as a negative regulator of Wnt signaling, controlling the production of mature DA neurons *in vivo* in zebrafish and in human stem cell-based neural differentiation assays. Taken together, these results suggest a conserved and crucial role for *miR-7* in regulating DA neurogenesis in the vertebrate brain.

## Figures and Tables

**Figure 1 cells-09-00711-f001:**
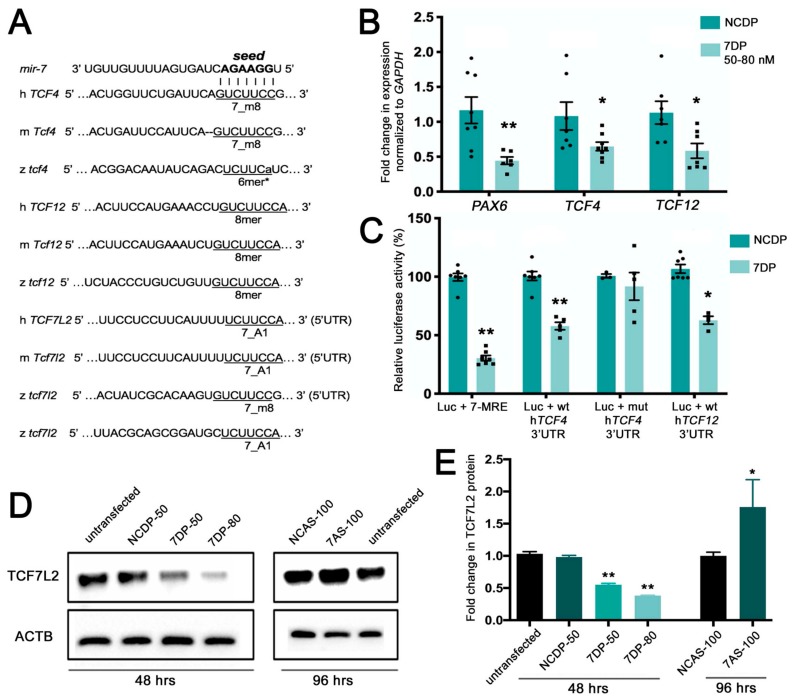
*miR-7* negatively regulates *TCF7L2,* encoding the Wnt/β-catenin transducer, and the basic helix–loop–helix (bHLH) genes *TCF4* and *TCF12.* (**A**) Various seed match regions (7_m8, 6mer*, or 8mer types) for *miR-7* are detectable in the human (h), mouse (m), and zebrafish (z) 3′UTRs of *Tcf4* and *Tcf12* transcripts. A seed match region for *miR-7* is detectable in the 5′UTR of *Tcf7l2* transcripts in the three species; an additional 7_A1 seed region is detectable in the 3′UTR of the zebrafish *tcf7l2* transcript. (**B**) *TCF4* and *TCF12* are down regulated by *miR-7,* following *miR-7* over-expression (*miR-7* duplex (7DP)). *PAX6* is used as a positive control being a known target of *miR-7*. Normalization was done to *GAPDH* and presented as fold change ± SEM (*n* ≥ 3) relative to the expression of control duplex NCDP; (*) *p* < 0.05 and (**) *p* < 0.01. (**C**) Luciferase reporter assay: the 3′UTR region of *TCF12* and the 3′UTR of b-fragment of *TCF4* were cloned downstream of the *Renilla* luciferase gene in the psiCheck2 vector. Repression of the luciferase activity was observed for *miR-7* miRNA response element (MRE) (75%), *TCF4* 3′UTR (40%), and *TCF12* 3′UTR (~40%). Every *Renilla* luciferase reading was normalized to that of the control firefly luciferase. The luciferase activities of *miR-7* transfected cells were presented as percentage relative to the level of luciferase in NCDP transfected cells (this control luciferase level is considered as 100%). The values represent average ± SEM (*n* ≥ 8). Two-tail t-test results are indicated by (**) *p* < 0.01 and (*) *p* < 0.05, relative to NCDP. (**D**) *miR-7* represses the endogenous TCF7L2 protein levels in neural progenitor cells 48 h after transfection with 7DP at two tested doses: 50 nM and 80 nM (control duplex: NCDP50). Antisense down-regulation of *miR-7* (7AS-100) leads to increased levels of TCF7L2 protein, compared with control antisense (NCAS-100) and untransfected controls (analysis at 96 h after transfection). (**E**) The TCF7L2 protein levels were quantified from the Western blot bands, normalized to the β-actin (ACTB) levels and presented as fold change ± SEM (n ≥ 3). Two tailed *t*-tests are indicated by (*) *p* < 0.05 and (**) *p* < 0.01.

**Figure 2 cells-09-00711-f002:**
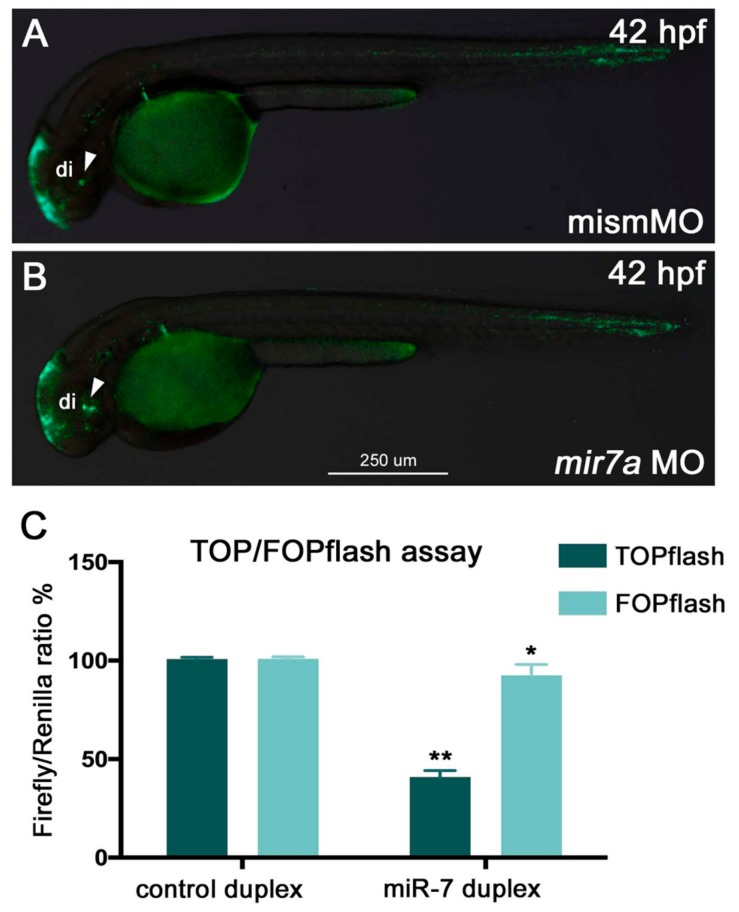
*miR-7* negatively regulates Wnt signaling in zebrafish embryos and HEK293T cells. (**A,B**) *miR-7* knockdown in zebrafish (*mir7a* MO) resulted in an increase of Wnt reporter expression in the ventral brain (**B**, green signals indicated by the white arrowhead) when compared with embryos injected with the control mismatch morpholino (mismMO) (**A**). Both panels display lateral views of 42 hpf *Wnt:GFP* transgenic embryos, anterior to the left; di: diencephalon. (**C**) HEK293T cells were co-transfected with TOPflash vector, NCDP, or 7DP, as well as *Renilla* luciferase control vector (pRL-TK). Similarly, the FOPflash vector (with mutated Tcf binding sites) was co-transfected with NCDP and 7DP, as well as pRL-TK. Luciferase activity was measured at 48 h post-transfection. About a 60% reduction in luciferase activity was observed in the TOPflash vector, whereas not much significant change was observed in the FOPflash vector. Luciferase readings were normalized to the co-transfected control *Renilla* luciferase vector pRL-TK plasmid. The error bars represent luciferase activity average ± SEM, where the experiment was done four times with two biological replicates each time (*n* = 6). Two-tailed *t*-tests are described as (**) where *p* < 0.01, and (*) where *p* < 0.05.

**Figure 3 cells-09-00711-f003:**
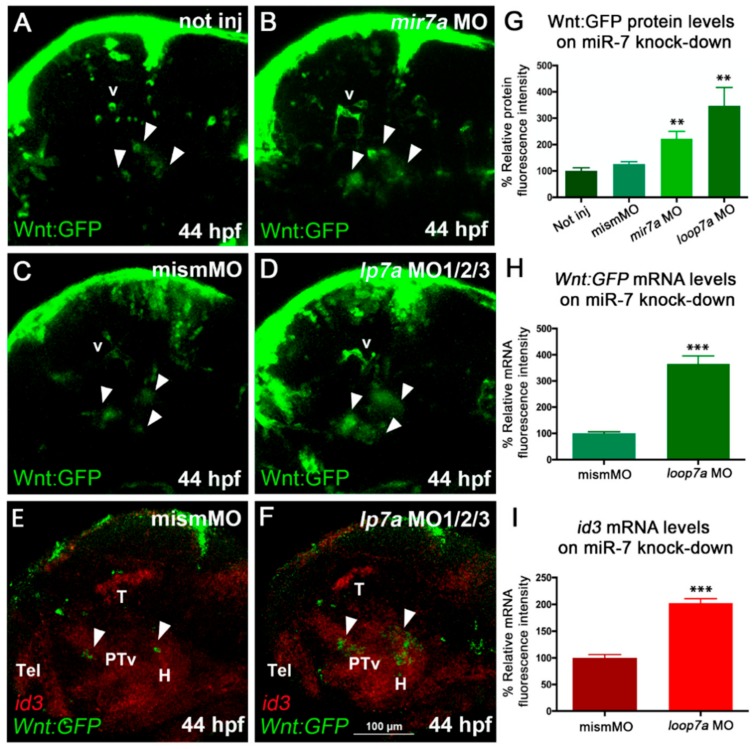
*miR-7* knockdown increases Wnt signaling in the zebrafish diencephalon. (**A**,**B**,**C**) Morpholino-mediated knock-down of *miR-7* (mature form) increases Wnt-reporter GFP (Green Fluorescent Protein) in the diencephalon (arrowheads) of *miR-7* morphants (**B**) compared with not injected (**A**) and mismatched MO-injected (**C**) controls. Increased Wnt-reporter GFP is confirmed in *loop7a1/2/3* (immature *miR-7*) morphants (**D**). (**E**,**F**) Two-color *in situ* hybridization for *id3* (landmark) and *Wnt:gfp* mRNAs shows that most of the diencephalic Wnt-responsive cells (arrowheads) are located in the ventral posterior tuberculum (PTv) and hypothalamic (**H**) regions of 44 hpf embryos. *Loop7a1/2/3* morphants display increased *id3* and *gfp* expression, compared with controls. All panels show 44 hpf embryonic heads in lateral view, from anterior to the left. Tel: telencephalon; T: tectum; v: vessels. (**G**,**H**,**I**) Quantitative analysis of Wnt reporter in the zebrafish ventral diencephalon. Chart G refers to A–D experimental series; charts H and I refer to E–F experiments. Error bars represent ± SEM. The experiments were repeated thrice. Sample size n = 5 for quantification using Volocity software. Unpaired *t*-test results are indicated by (**) *p* < 0.01 and (***) *p* < 0.001. All panels zoom on the dorsal diencephalic (dd) region of 2 dpf embryos in lateral view, from anterior to the left. Displayed images represent the average phenotype from batches of *n* = 20 embryos per condition; the experiment was performed in duplicate.

**Figure 4 cells-09-00711-f004:**
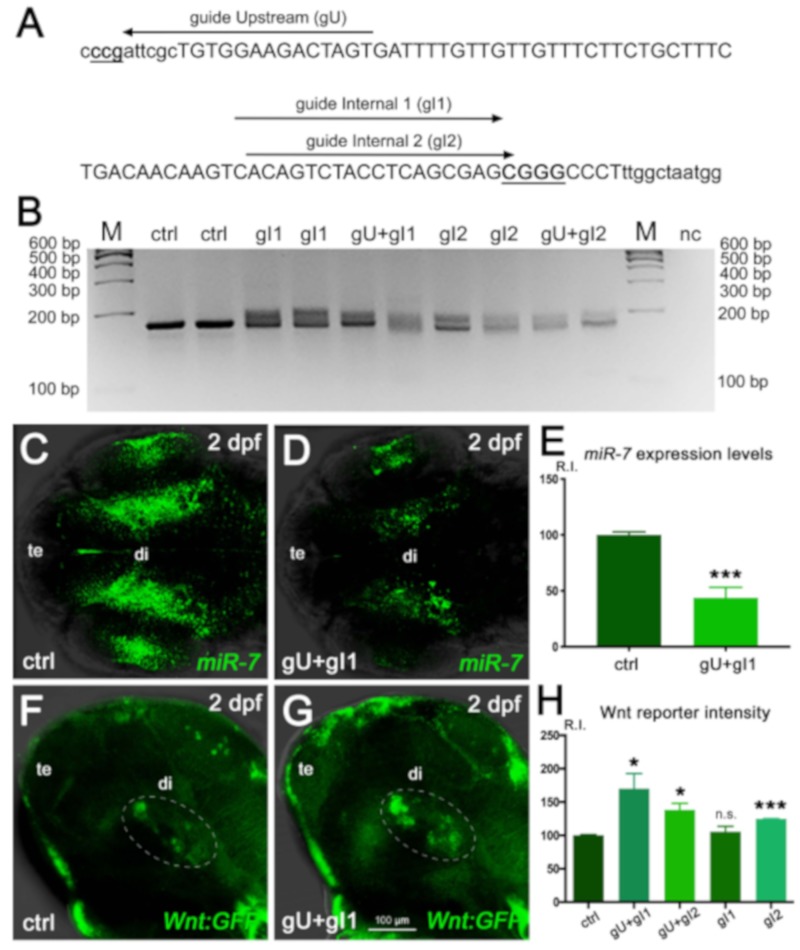
*miR-7* gene editing decreases *miR-7* expression and increases Wnt signaling in the zebrafish diencephalon. (**A**) Position of the three Crispr/Cas9 guides used to perform gene editing in the zebrafish *miR-7a* locus. (**B**) Agarose gel electrophoretic analysis of the Crispr/Cas9-targeted region in the *miR-7a* gene, PCR-amplified from genomic DNA of control (ctrl) or guide-injected (g) embryos. Smears indicate Crispr/Cas9-induced gene modification. M: size marker; nc: PCR negative control. (**C**–**E**) Decreased *miR-7* expression is observed in crispants injected with the gU + gI1 Crispr/Cas9 guides, compared with the control (ctrl). Signal quantification (R.I.: relative intensity) is shown in **E**; (***), *p* < 0.001, *n* = 6. (**F**–**H**) Analysis of diencephalic Wnt signaling (green cells within the white dashed ellipse) in the control (ctrl, F) and guide-injected (gU + gI1, G) Wnt:GFP transgenic embryos (**G**). Signal quantification is shown in (**H**). The highest Wnt signal increase (80% more than the control level) is reached with the gU + gI1 guide combination; (*) *p* < 0.05 and (***) *p* < 0001; (n.s.), not significant; *n* = 6. All embryonic heads are at 2 dpf and in dorsal (**C**,**D**) or lateral (**F**,**G**) view, with anterior to the left. te: telencephalon; di: diencephalon.

**Figure 5 cells-09-00711-f005:**
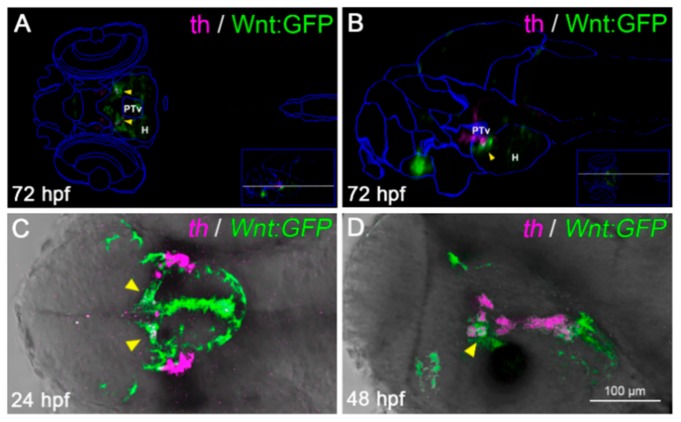
A subset of diencephalic Wnt-responsive cells express the dopaminergic (DA) marker tyrosine hydroxylase (TH). (**A**,**B**) Virtual anatomical reconstruction of a 72 hpf zebrafish brain, generated by ViBE-Z software, merging Wnt:GFP (green) and tyrosine hydroxylase (*th*, magenta) expression domains, compared with reference anatomical areas (blue lines). Virtual co-localizations (yellow arrowheads) are detected at the boundary between the ventral posterior tuberculum (PTv) and hypothalamus (H). (**A**,**B**) are dorsal and lateral views, respectively, with the anterior to the left, across the PTv/H boundary (section planes shown by white lines in the insets). (**C**,**D**) Co-localization of *th*-positive and Wnt-responsive areas (yellow arrowheads) is confirmed by dual color *in situ* hybridization in 24 and 48 hpf embryos. (**C**,**D**) are dorsal and lateral head views, respectively, with the anterior to the left.

**Figure 6 cells-09-00711-f006:**
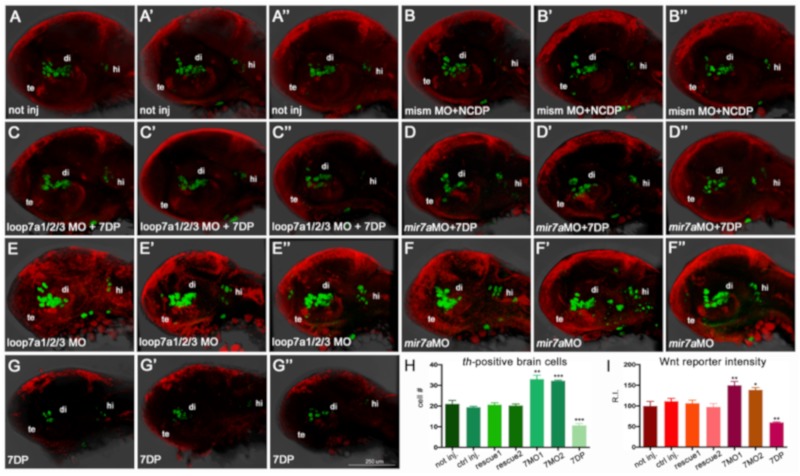
*miR-7a* negatively regulates Wnt signaling and diencephalic *th*-positive cells in the zebrafish brain (**A**–**G’’**): The knock-down of *miR-7a* activity by two independent morpholinos, targeting the immature and mature forms (*loop7a1/2/3* MO, shown in **E**,**E’**,**E’’**, and *mir7a* MO, shown in **F**,**F’**,**F’’**) increases the amount of *th*-positive brain cells (in green) and the activity of a Wnt reporter (in red), compared with controls not injected (**A**,**A’**,**A’’**), co-injected with non-functional morpholino (mismMO) + non-functional *miR-7a* (NCDP) (**B**,**B’**,**B’’**), or co-injected (rescued) either with *loop7a1/2/3* MO + 7DP (**C**,**C’**,**C’’**) or with *miR-7a* MO + 7DP (**D**,**D’**,**D’’**). On the contrary, over-expression of 7DP reduces the amount of *th*-positive brain cells and the activity of the Wnt reporter (**G**,**G’**,**G’’**). All pictures show the head region of 40 hpf embryos in lateral view, with the anterior to the left. te: telencephalon; di: diencephalon; hi: hindbrain. (**H**,**I**): Charts showing the *th*-positive cell counting (**H**), and the measure of Wnt reporter activity (**I**), expressed as fluorescence relative intensity (R.I.), in the seven conditions. Sample size *n* = 5 measures/condition; (*) *p* < 0.05; (**) *p* < 0.01; (***) *p* < 0.001.

**Figure 7 cells-09-00711-f007:**
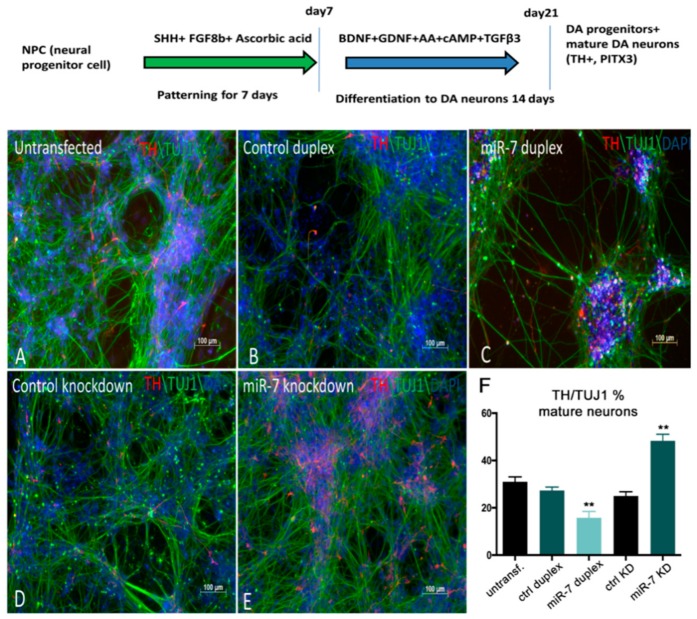
*miR-7* negatively regulates the number of TH+ DA neurons derived from H9NPC cells. (**A**,**B**,**C**) Control H9NPC cells (**A**) showing TH staining at day 10 of DA neurogenesis (schematized in the top of the figure). BDNF: Brain Derived Neurotrophic Factor; GDNF: Glial cell Derived Neurotrophic Factor; cAMP: cyclic Adenosine MonoPhosphate; TGFβ3: Transforming Growth Factor beta 3. Decrease of TH+ cells (red) was observed when cells were transfected with *miR-7* duplex (**C**), compared with the control duplex (**B**) and untransfected cells (**A**). (**D**,**E**) Increase of TH+ cells was observed upon knockdown of miR-7 (**E**), when compared with the control antisense. (**F**) Ratio of TH+/TUJ1 cells was quantified by manual counting of the neurons from images captured from 10 different areas on a random basis. Error bars represent average% ± SEM; the experiment was repeated thrice with *n* = 3 different biological replicates; (**) *p* < 0.01.

**Figure 8 cells-09-00711-f008:**
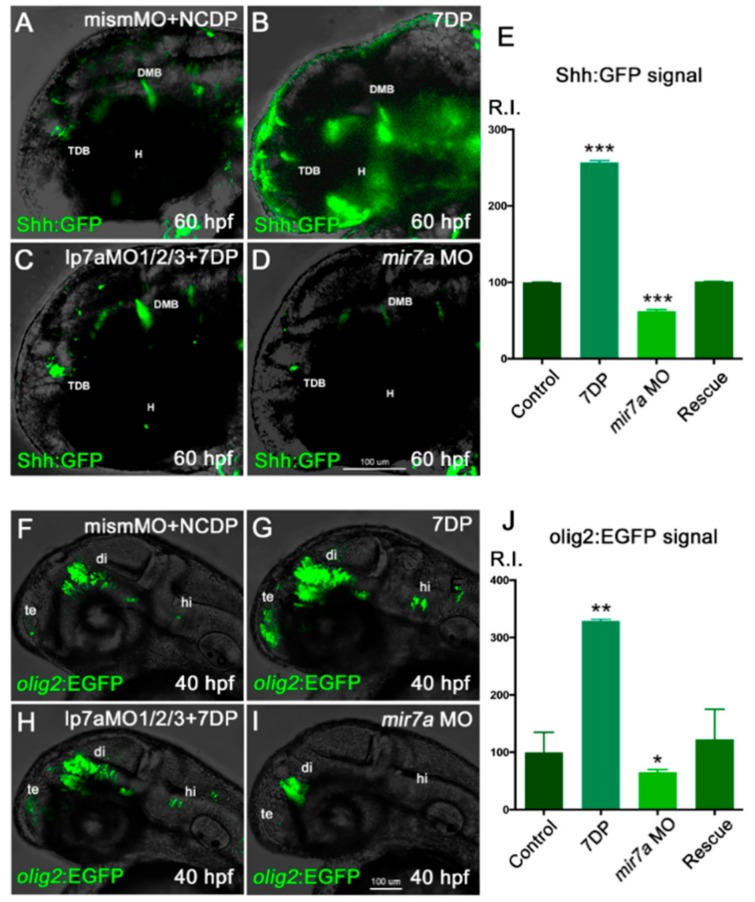
*miR-7* positively regulates Shh signaling responsiveness and *olig2*+ cells in the zebrafish diencephalon. (**A**–**D**) *Shh:GFP* reporter embryos, injected with NCDP and mismatch morpholino, were used as controls (**A**). Injection of *miR-7* 7DP (**B**) or *miR-7a* morpholino alone (**D**) elicits opposite effects on Shh-responsive cells located in the diencephalic region, between the telencephalic-diencephalic boundary (TDB), the diencephalic-mesencephalic boundary (DMB), and the hypothalamus (H), compared with injected controls (**A**). Co-injection of 7DP and *miR-7a* MO rescues both conditions (**C**). All panels show the head region of 60 hpf embryos in lateral view, with the anterior to the left. (**E**) Chart refers to A–D experimental series. Error bars represent SEM; the experiments were repeated thrice. Sample size n = 5 measures/condition for quantification using Volocity software; (***) *p* < 0.001. (**F**–**I**) *olig2:EGFP* embryos, injected with NCDP and mismatch morpholino, were used as controls (**F**). Telencephalic (te), diencephalic (di) and hindbrain (hi) expression of *olig2*-dependent EGFP appears increased in embryos overexpressing *miR-7* 7DP (**G**), and reduced in *miR-7a* morphants (**I**). Co-injection of *miR-7a* MO and 7DP rescues the phenotype (**H**). All panels are lateral views of the head region at 40 hpf, with the anterior to the left. (**J**) Chart refers to F–I experimental series. Error bars represent SEM; the experiments were repeated thrice. Sample size *n* = 5 measures/condition for quantification using Volocity software; (*) *p* < 0.05; (**) *p* < 0.01.

**Figure 9 cells-09-00711-f009:**
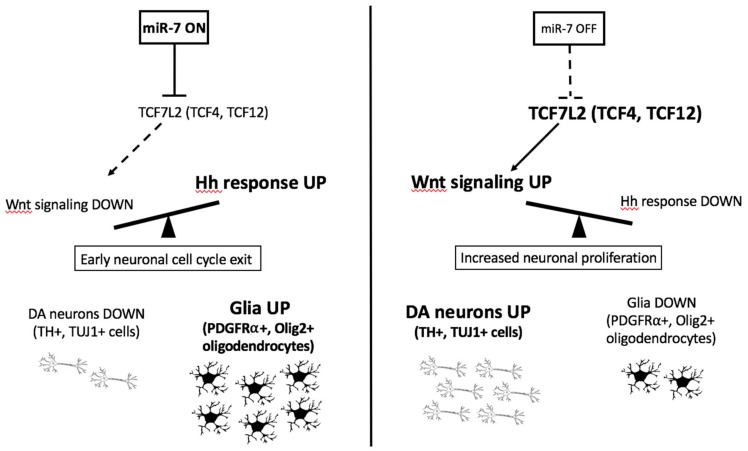
*miR-7* regulates Wnt/Shh responsiveness and the balance between DA neurons and glia. The model summarizes the *in vivo* (zebrafish) and *in vitro* (human cells) findings from this work on *miR-7* function, showing the direct regulation of Wnt signaling (by the Wnt-transducer TCF7L2) and possibly indirect regulation of Shh responsiveness, with consequences on the relative amount of DA neurons and glial cells generated during neural progenitor differentiation.

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
