# Peer review of "miR-7 Controls the Dopaminergic/Oligodendroglial Fate through Wnt/β-catenin Signaling Regulation"

_cells, 2020, doi:10.3390/cells9030711_

Round 1

Reviewer 1 Report

In the manuscript "miR-7 controls the dopamine/oligodendroglial fate through Wnt/β-catenin signalling regulation", Adusumilli et al. describe the function of the miR-7 in neurogenesis in human tissue culture and zebrafish. The role of miR-7 is well studied since its discovery in 2001, and it has been suggested that it plays crucial roles in both organ development and biological functions of cells. Furthermore, miR-7 plays a vital role in suppressing the migration, colony formation, and cell-cycle progression in cancers. In the nervous system, miR-7 is expressed in neurons and acts as a repressor of α-Syn protein level. In Parkinson's mouse models, the miR-7 expression is decreased in DA neurons.

Here, the authors show that miR-7 is specifically expressed in a subset of neurons, which give rise to TH positive neurons - most likely DA neurons. Furthermore, the authors show that these neurons co-localize with cells active for the Wnt/β-catenin pathway. Next, the authors provide evidence that miR-7 interferes with Wnt/β-catenin signalling and as a consequence, the miR-7 limits the DA neuronal populations by inhibiting Wnt/β-catenin signalling. In the last part of the work, the authors suggest that miR-7 influences the interaction between Wnt/β-catenin signalling and Shh signalling. In general, this is a good manuscript as it combines nicely in vitro and in vivo work. Furthermore, the authors identify a novel role for miR-7 in DA development.

However, based on the experiments provided, the authors draw some conclusions which require further support. The link between miR-7 function / DA neuronal development / Wnt/β-catenin signalling is provided only indirectly. I would suggest adding more experiments to support this conclusion. Furthermore, the authors have generated a CRISPR-based mutant of miR-7, and it is unclear to me why this tool has not been used more often in the analysis.

Specific comments

Line 203: The material and methods section is well written and contains many useful details. The authors mention that the experiments were done three times and in triplicates. How did the authors ensure independent measurements? This needs to be explained.  

The statistical analysis in Figure 1 is unclear - e.g. the error bars in b are extremely short - however, the significance is only p<0.05. It is unclear how often the experiments were done and what is the actual spread of the data. I would recommend using box plots for illustrations.

Line 388: The authors claim that the expression is detectable at 36hpf - however, this is not shown in the Figure.

The expression/ISH analysis at 3years is impressive.

Figure 2: IMO, the illustration of the 7xTCF-Xla.Siam:GFP reporter line is not necessary as it was published in 2012 already. I would prefer to move the CRISPR data from the supplementary part to the main figures instead - as these results are much more important.

Figure 3: It is unclear which fluorescent area in the brain has been quantified.

Figure 4: ViBE-Z is a powerful platform to compare gene expression on a tissue level. However, double staining of Wnt positive cells together with TH cells, would be more appropriate allowing a single-cell resolution.   

The link between miR-7 and Wnt signalling and DA development is exciting and needs to be strengthened. I would strongly suggest adding a set of experiments in which Wnt levels are altered in combination with functional interference with miR-7. The development of the DA neurons could be used as a read-out. For example, there are numerous, chemical activators and inhibitors of the Wnt pathway, which could be used to "rescue" the DA phenotype after overexpression or knock-down of miR-7. This set of experiments could be done in zebrafish and/or in H9NPC cells. This would be a strong argument to support the conclusion that miR-7 regulates DA development via altering the Wnt/β-catenin signalling pathway. 

A similar point needs to be shown for Shh signalling - if the authors want to extend their conclusion to the Hh field. To conclude a functional interaction solely based on co-localisation and alteration in a reporter system is not sufficient.

Author Response

Reviewer 1 Comments and Suggestions for Authors

In the manuscript "miR-7 controls the dopamine/oligodendroglial fate through Wnt/β-catenin signalling regulation", Adusumilli et al. describe the function of the miR-7 in neurogenesis in human tissue culture and zebrafish. The role of miR-7 is well studied since its discovery in 2001, and it has been suggested that it plays crucial roles in both organ development and biological functions of cells. Furthermore, miR-7 plays a vital role in suppressing the migration, colony formation, and cell-cycle progression in cancers. In the nervous system, miR-7 is expressed in neurons and acts as a repressor of α-Syn protein level. In Parkinson's mouse models, the miR-7 expression is decreased in DA neurons.

Here, the authors show that miR-7 is specifically expressed in a subset of neurons, which give rise to TH positive neurons - most likely DA neurons. Furthermore, the authors show that these neurons co-localize with

cells active for the Wnt/β-catenin pathway. Next, the authors provide evidence that miR-7 interferes with Wnt/

β-catenin signalling and as a consequence, the miR-7 limits the DA neuronal populations by inhibiting Wnt/β-

catenin signalling. In the last part of the work, the authors suggest that miR-7 influences the interaction between Wnt/β-catenin signalling and Shh signalling. In general, this is a good manuscript as it combines nicely in vitro and in vivo work. Furthermore, the authors identify a novel role for miR-7 in DA development.

However, based on the experiments provided, the authors draw some conclusions which require further

support. The link between miR-7 function / DA neuronal development / Wnt/β-catenin signalling is provided only indirectly. I would suggest adding more experiments to support this conclusion. Furthermore, the authors have generated a CRISPR-based mutant of miR-7, and it is unclear to me why this tool has not been used more often in the analysis.

We thank Reviewer 1 for his/her very accurate revision of our manuscript. The analysis of mir-7 started in the framework of a collaboration, initiated in Singapore using an in vitro setup and then continued in Padova (Italy) where in vivo experiments, performed in zebrafish, were allowed by the UniPD Ethical Committee only under transient conditions, within 1 week post-fertilization. Under these ethical constraints, all experiments were performed using antisense morpholino oligomers, being a strategy able to rapidly knock-down entire batches of embryos (differently from stable KO individuals, obtainable in Mendelian ratios), and easily tunable/combinable with over-expression studies and chemical treatments. In the last part of this project, Crispr/Cas9 strategies were applied solely under transient conditions, to confirm morpholino-based experiments, at embryonic stages not protected by the EU Directive 2010/63/EU. We are currently applying for a new ethical permission, to extend our analysis of miR-7 KO fish at adult stages.

Concerning the request for additional experiments to support our conclusions, these are described in the following sections.

We thank Reviewer 1 for his/her observations as we believe that these additional experiments have very much improved the original version of our paper.

Specific comments

Line 203: The material and methods section is well written and contains many useful details. The authors

mention that the experiments were done three times and in triplicates. How did the authors ensure independent

measurements? This needs to be explained.

We thank Reviewer 1 for giving us the opportunity to make this point clearer. For in vitro studies, performed in Singapore, each experiment was performed by thawing each time a new batch of cells (new passage number), transfecting them with the vectors, and performing cell lyses/luciferase assays in three technical replicates. For in vivo studies, carried out in Padova, embryos for each analyzed condition derived from three different injection sessions, and measurements were performed by combining results from three independent whole-mount in situ hybridization or immune-histochemical experiments. Importantly, after embryo collection from multiple crosses, individuals were randomized to avoid that differences could arise from different parental background. Also, not injected and control-injected individuals were always co-processed and co-stained in the same tube together with knock-down/over-expressed embryos, to ensure that differences in signal intensity could solely depend on the investigated condition and not on operator, protocol performances or type of tube.

The statistical analysis in Figure 1 is unclear - e.g. the error bars in b are extremely short - however, the

significance is only p<0.05. It is unclear how often the experiments were done and what is the actual spread of the data. I would recommend using box plots for illustrations.

We thank reviewer 1 for his/her suggestion; scatter plots have been added in Figure 1 to show the spread of the data.

Line 388: The authors claim that the expression is detectable at 36 hpf - however, this is not shown in the

Figure. The expression/ISH analysis at 3years is impressive.

We thank Reviewer 1 for spotting this inaccuracy; initial expression at 36 hpf was indeed observed during in situ hybridization experiments but, as correctly pointed out by Reviewer 1, not included in the figure. We have modified the text to clarify which data were communicated (and not shown) and which ones were illustrated in the figure.

Figure 2: IMO, the illustration of the 7xTCFXla. Siam:GFP reporter line is not necessary as it was published in 2012 already. I would prefer to move the CRISPR data from the supplementary part to the main figures instead - as these results are much more important.

We thank Reviewer 1 for this suggestion. The schema of the reporter line has been removed from Figure 2. Moreover, Crispr/Cas9 data have been moved from Supplementary Section to Main Figures. The figure on Crispr/Cas9 data is now the new Figure 4. The following figures have been renumbered accordingly.

Figure 3: It is unclear which fluorescent area in the brain has been quantified.

We thank Reviewer 1 for this specification that was inadvertently omitted. Measurements were performed on Wnt signals arising from the ventral diencephalon (ventral posterior tuberculum and hypothalamus); this information has been now added in the legend of Figure 3.

Figure 4: ViBE-Z is a powerful platform to compare gene expression on a tissue level. However, double staining

of Wnt positive cells together with TH cells, would be more appropriate allowing a single-cell resolution.

We fully agree with Reviewer 1 that the ViBE-Z platform alone has not single-cell resolution to clarify possible marker overlaps. In Figure 4 we have provided a “real” experiment, represented by a whole mount in situ hybridization, showing that a subset of TH cells is also Wnt reporter-positive.

The link between miR-7 and Wnt signalling and DA development is exciting and needs to be strengthened. I

would strongly suggest adding a set of experiments in which Wnt levels are altered in combination with functional interference with miR-7. The development of the DA neurons could be used as a read-out. For example, there are numerous, chemical activators and inhibitors of the Wnt pathway, which could be used to "rescue" the DA phenotype after overexpression or knock-down of miR-7. This set of experiments could be done in zebrafish and/or in H9NPC cells. This would be a strong argument to support the conclusion that miR-7 regulates DA development via altering the Wnt/β-catenin signalling pathway.

A similar point needs to be shown for Shh signalling – if the authors want to extend their conclusion to the Hh

field. To conclude a functional interaction solely based on co-localisation and alteration in a reporter system is

not sufficient.

We thank Reviewer 1 for his/her very helpful suggestions, that contributed to a net improvement of the whole study. We have performed a pharmacological modification of Wnt and Shh signaling in combination with the manipulation of miR-7 levels. We are happy to anticipate that the results are supporting previous conclusions; these analyses are presented in two supplementary figures (Suppl. Fig. 9 and Suppl. Fig. 10).

Reviewer 2 Report

The paper by L. Adusumilli et al. entitled “miR-7 controls the dopaminergic/oligodendroglial fate through Wnt/β-catenin signaling regulation” appears to have some relevance since the great interest concerning the use of a microRNA approach. The authors focused their attention on microRNA-7 (miR-7), whose regulation of cell signaling and neural development is still unclear. They studied the role of miR-7 by in vitro and in vivo models consisting on human stem cell cultures and whole zebrafish embryos respectively, in the development of dopaminergic (DA) neurons highly involved in Parkinson’s disease. In particular, Adusumilli et al. studied the modulation, by miR-7, of different transcription factors and signaling cascades, such as the Wnt and Shh pathways. In particular, the authors observed that miR-7: i) is expressed in the forebrain during development of DA neurons in zebrafish; ii) downregulates 143 target genes including the neural fate markers TCF4 and TCF12, and the TCF/LEF effector of the Wnt pathway TCF7L2; iii) regulates negatively the proliferation of DA-progenitors by inhibiting Wnt/β-catenin signaling in zebrafish embryos; and iv) regulates positively Shh signaling to control the balance between oligodendroglial and DA neuronal cell fate. Collectively, these results seem to suggest a new mechanism of crosstalk between Wnt and Shh signaling pathways during development of DA-neurons, which is mediated by a microRNA and that could represent a promising target in cell differentiation-based therapies for Parkinson’s disease.

Although the paper is quite interesting given the importance of the development of innovative therapeutic strategies against Parkinson’s disease, there are several concerns about this paper.

Major Point

In the drafting of the manuscript the authors have not followed a homogeneous form of writing. In particular, throughout the text the authors wrote the same words in different ways using, sometime the words in extended or abbreviations for the same words (mir-7 duplex instead of 7DP …..). This creates confusion in the reader.

Sometimes the reading of the present paper is not easy. The authors should try to streamline the text where it is possible, even using tables or schemes

Figures 1, 2 and 3 should include internal controls in the absence of control duplexes.

Author Response

Reviewer 2 Comments and Suggestions for Authors

The paper by L. Adusumilli et al. entitled “miR-7 controls the dopaminergic/oligodendroglial fate through Wnt/β-

catenin signaling regulation” appears to have some relevance since the great interest concerning the use of

a microRNA approach. The authors focused their attention on microRNA-7 (miR-7), whose regulation of

cell signaling and neural development is still unclear. They studied the role of miR-7 by in vitro and in vivo

models consisting on human stem cell cultures and whole zebrafish embryos respectively, in the development of dopaminergic (DA) neurons highly involved in Parkinson’s disease. In particular, Adusumilli et al. studied the modulation, by miR-7, of different transcription factors and signaling cascades, such as the Wnt and Shh pathways. In particular, the authors observed that miR-7: i) is expressed in the forebrain during development of DA neurons in zebrafish; ii) downregulates 143 target genes including the neural fate markers TCF4 and TCF12, and the TCF/LEF effector of the Wnt pathway TCF7L2; iii) regulates negatively the proliferation of DA-progenitors by inhibiting Wnt/β-catenin signaling in zebrafish embryos; and iv) regulates positively Shh signaling to control the balance between oligodendroglial and DA neuronal cell fate. Collectively, these results seem to suggest a new mechanism of crosstalk between Wnt and Shh signaling pathways during development of DA-neurons, which is mediated by a microRNA and that could represent a promising target in cell differentiation-based therapies for Parkinson’s disease.

Although the paper is quite interesting given the importance of the development of innovative therapeutic strategies against Parkinson’s disease, there are several concerns about this paper.

Major Point

In the drafting of the manuscript the authors have not followed a homogeneous form of writing. In particular, throughout the text the authors wrote the same words in different ways using, sometime the words in extended or abbreviations for the same words (mir-7 duplex instead of 7DP …..). This creates confusion in the reader.

We thank Reviewer 2 for his/her helpful comments and for spotting the inaccuracy in the use of the abbreviations. We have now re-checked the whole manuscript and modified the critical parts to have a more homogeneous use of the 7DP and NCDP abbreviations. All changes are highlighted in yellow.

Sometimes the reading of the present paper is not easy. The authors should try to streamline the text where it is possible, even using tables or schemes

We thank Reviewer 2 for his/her comments; wherever possible, the text has been shortened/simplified, as well as revised by a US colleague, as no one of the authors is native English speaker. We apologize if the previous version was not sufficiently fluent; we hope that this revised version may now satisfy readers’ expectations.

Figures 1, 2 and 3 should include internal controls in the absence of control duplexes.

We have checked Figures 1, 2 and 3 in relation to this important point, raised by Reviewer 2. In Figure 1 and 2, the charts refer to in vitro experiments performed in the Singapore lab at the beginning of this study. Unfortunately, that lab ceased its activity soon after the end of this research, so those experiments currently maintain as controls only control duplex (NCDP) conditions. Indeed, it will be difficult to replicate these in vitro experiments somewhere else, within the time frame allowed by the journal for the revision (1 month). However, as shown in the E chart, untransfected cells were effectively compared with NCDP transfected cells, at least for Tcf7l2 and Actb levels, not detecting significant differences between conditions. Concerning Figure 3 and, in general, all zebrafish-based experiments using NCDP, a cumulative table (Suppl. Table 1), included in the paper (Supplementary Section), can be taken into consideration to verify the negligible effect of NCDP, compared to not injected controls. The remaining figures of the paper, as noted by the reviewer, are instead all accurately distinguishing between not injected and injected controls, never detecting any significant difference between these conditions. As far as concerns the revision experiments and related figures, the “ctrl” (control) condition indicates imaging and measurements performed in both not injected and injected controls, considered as a single sample population.

Round 2

Reviewer 2 Report

The authors, according to the referee’s suggestions, have answered that the referees had requested. The revised version of the paper entitled “miR-7 controls the dopaminergic/oligodendroglial fate through Wnt/β-catenin signaling regulation” by Lavanya Adusumilli, Nicola Facchinello, Cathleen Teh, Giorgia Busolin, Minh Thi Nguyet Le, Henry Yang, Giorgia Beffagna, Stefano Campanaro, Wai Leong Tam, Francesco Argenton, Bing Lim, Vladimir Korzh, Natascia Tiso, is now suitable for publication.